# The Role of Cross-Correlations in the Multi-Tracer Area

**Alain Blanchard** 

IRAP, OMP, UPS, Université de Toulouse, CNRS, 14 Avenue Edouard Belin, F-31400 Toulouse, France; alain.blanchard@irap.omp.eu; Tel.: +33-05-61-33-28-42

**Abstract:** Mapping the same volume of space with different tracers allows us to obtain information through estimated quantities exploiting the multi-tracer technique. Indeed, the cross-correlation of different probes provides information that cannot be otherwise obtained. In addition, some estimated quantities are not sensitive to the noise produced by the sampling variance but are only limited by the shot (or Poisson) noise, an attractive perspective. A simple example is the ratio between the (cross)-correlations, measuring the ratio of the bias parameters. Multi-tracer approaches can thereby provide additional information that cannot be extracted from independent volumes.

**Keywords:** cosmology; galaxy surveys; cosmological parameters

## 1. Introduction

Cosmology has undergone remarkable progress over the last decades. The concept of inflation has triggered the idea that the initial state of the Universe as well as its contents are set up in the very early times, at energy far beyond what can be directly reached by human-made technology: relevant energy scales are anticipated to be of the order of $10^{15}$ GeV while LHC is investigating physics at a maximum energy of the order of $10^4$ GeV. The perspective for direct investigations seems therefore hopeless. However, the fluctuations necessary for structure formation probably emerged at those energies and most of their later evolution is within the linear regime, i.e., their amplitude remains small during their history. Two major observables are accessible to astronomy to diagnose these fluctuations: the cosmic background radiation, essentially a picture of the universe at $z \sim 1100$, and the matter distribution at "low redshift", i.e., redshift from 0 to 5, today, which may be extended up to something like 30 in the future. While early observations were performed over a limited area of the sky, it has been realized that the ultimate limitations come from the limited size of the sampled volumes. Concerning the CMB intensity, Planck has almost closed the possibility to gain significant further information, despite the fact that there is still room for improvements from small-scale measurements. For galaxy surveys, on the other hand, there is much more to gain [1,2].

## 2. The Period of Cosmological Surveys

In the 1990s, the need for large volume surveys triggered the emergence of several large-scale surveys of the CMB, whose apotheosis were WMAP and Planck, while the concept of dedicated optical telescopes for large galaxy surveys with the SDSS, the 2dF, the Hobby–Eberly telescope were a few examples of these early projects. Today, Euclid and LSST are the most emblematic examples of this strategy which is likely to be pursued in future decades.

The use of cosmological constraints obtained from different probes was early identified as a powerful way to achieve tighter constraints essentially because one probe is generally sensitive to a combination of different cosmological parameters rather than to a unique one and the combination being specific to each probe. The advancement of several new large-scale surveys holds the promise of much better results in combination than those obtained by individual surveys [3]. Not only different probes might measure different

combinations of cosmological parameters, but they are likely to be sensitive to different systematic errors.

### 3. Using Cross-Correlation on Galaxy Samples

However, when two probes are measured within the same volume, a new data vector can be built from the cross-correlation between the two probes which provides obviously additional information. The cross-correlation between a CMB map and a galaxy sample was early identified as a possible diagnostic of the late behavior of the expansion.

Galaxy surveys are by now limited by the so-called sampling variance, sometimes referred to as cosmic variance[1]. In order to use the cross-correlation between two tracers it is obviously necessary that the volumes on which each tracer has been surveyed must overlap. This was unlikely when the sizes of surveys were modest. However, with the emergence of the biased picture in which galaxies provided of a biased picture of the dark matter density field, the question arose whether luminous galaxies were more biased than fainter ones. This was a generic property expected in scenarios in which the galaxy population was strongly biased. However, a direct comparison of the correlation function from bright and faint galaxies in a given (magnitude-limited) survey was leading to ambiguous results because the surveyed volumes were of different depths. The method of multi-tracers was first developed by using the ratio of correlations and cross-correlations to address this issue [4], showing that luminous galaxies are more clustered than faint galaxies in a statistically significant way, despite the fact that the samples used were from a few hundred galaxies!

The use of combination and of cross-correlations is well illustrated by the forecasts performed for the cosmological parameters that will be extracted from the Euclid surveys. The two main probes are the weak-lensing (WL) signal extracted from the photometric sample and from the spectroscopic survey (GCs). However, the clustering of galaxies within the photometric sample is an additional source of information (GCph). On its own, its efficiency, measured, for instance, by the Figure of Merit (FoM), is rather poor (FoM < 10). However, when combined with WL or with GCs, it boosts significantly the FoM. Furthermore, the addition of cross-correlation (Xc) between the WL and GCp leads to an additional boost allowing the FoM to reach the requirement (FoM > 400), even in the most pessimistic configuration [5], as the boost on the FoM can be of the order of 3 to 5. This is illustrated by Figure 1 where the role of cross-correlations is enlightened [5]. The improvements provided by the addition of cross-correlation were further examined in detail by [6] and confirmed their relevance for many parameters.

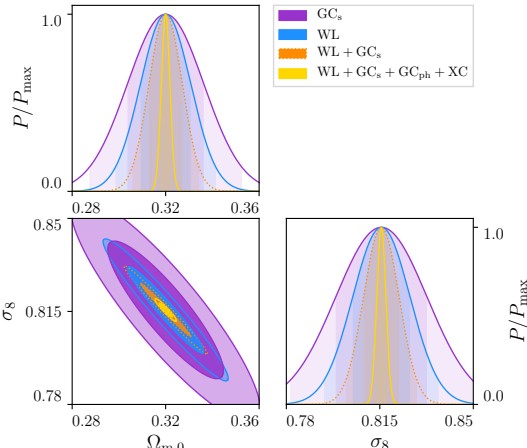

**Figure 1.** This picture illustrates the role of including cross-correlations when possible. These are the forecasts on the constraints expected from the Euclid space mission for a CPL flat cosmology, for the density parameter $\Omega_{m,0}$ and the amplitude of matter fluctuations $\sigma_8$ for spectroscopic sample (purple), weak-lensing (blue), their combination (orange), and with the addition of the cross-correlation between the photometric sample and the weak-lensing (yellow), currently denoted $3 \times 2$ pt. From [5].

### 4. The Gain from Multi-Tracer Surveys

The interest of multi-tracer surveys, i.e., surveys that sample several tracers over overlapping volumes of the universe has greatly increased in the recent years [7]. The fundamental reason is that relative bias of distinct tracers can be estimated by estimators which are not limited by the sampling variance. This stems from the fact that if we have two populations $A$ and $B$, tracing the same density field of dark matter $\delta$, then $\delta_A = \delta N_A / N_A$ traces $b_A \delta$ and $\delta_B = \delta N_B / N_B$ traces $b_B \delta$, e ratio:

$$\left\langle \frac{\delta_A}{\delta_B} \right\rangle = \frac{b_A}{b_B} \tag{1}$$

does not contain the density term $\delta$ and is therefore not sensitive to the sampling variance.

This is the heart of the cross-correlation method [4,8] and has also been formulated in terms of the power spectrum [9,10]. This method can be applied to the measurement of non-Gaussianity $f_{NL}$ [11] and to redshift space distortion (RSD) [12].

Let us give a short insight into the origin of the suppression of the sampling noise in the estimation of the ratio of cross-correlations. For this, let us consider two populations $A$ and $B$, sampling a volume $V$ with a known selection function, allowing to determine their number density $n_A$ and $n_B$. Note that, in general, the density $n$ of a tracer is not given by $\tilde{n} = N/V$ ($N$ being the total number of tracers in the volume $V$). Using $\tilde{n}$ may bias some estimations of clustering quantities because of the integral constraint.

Let us denote by $i$ the label of the population $A$, i.e., $i$ goes from 1 to $N_A$. The volume of the shell between $r$ and $r + dr$ centered on tracer $i$ is $dV_i$, and $dN_i^B$ is the number of neighbors of tracers $B$ in this shell. Considering tracer $i$, the dark matter contrast in the shell centered on this tracer is $\Delta_i$. The number of neighbors $dN_i^B$ is the Poisson realization of the mean $ndV_i(1 + b_{AB}\Delta_i)$, which is the statistical mean over population $A$ and samples:

$$\left\langle dN_i^B \right\rangle = n_B \langle dV_i \rangle (1 + \xi_{AB}(r)) = n_B \langle dV_i \rangle (1 + b_A b_B \xi(r)) \tag{2}$$

Identically, for the number $dN_i^A$ of tracers $A$, we have:

$$\left\langle dN_i^A \right\rangle = n_A \langle dV_i \rangle (1 + \xi_{AA}(r)) = n_B \langle dV_i \rangle (1 + b_A^2 \xi(r)) \tag{3}$$

where

$$\langle \Delta_i \rangle = \xi(r) \tag{4}$$

is the correlation function of dark matter. Relation (2) stems from the (unbiased) estimator $\hat{\xi}_{AB}$ of $\xi_{AB}$:

$$\hat{\xi}_{AB} = \frac{\sum_i (dN_i^B - n_B dV_i)}{\sum_i n_B dV_i} \tag{5}$$

We can therefore write down an estimator of the ratio of the bias:

$$\widehat{\left(\frac{b_B}{b_A}\right)} = \frac{n_A}{n_B} \frac{\sum_i (dN_i^B - n_B dV_i)}{\sum_i (dN_i^A - n_A dV_i)} \tag{6}$$

The variance of this estimator (over different Poisson realizations of population $B$) is coming only from the shot (Poisson) noise of the $dN_i^B$. So, the variance of the above estimator is:

$$\left(\frac{n_A}{n_B}\right)^2 \frac{1 + \xi_{AB}}{\xi_{AB}^2} \frac{1}{N_B} \tag{7}$$

where $N_B = \sum_i n_B dV_i$. It is clear that this variance is tending towards zero when $N_B$ is tending to infinity and is therefore free of the sampling variance.

*A Final Remark*

The estimator of the correlation function (5) needs the knowledge of the number density of the tracers $n_B$. This quantity has to be estimated from the luminosity function (not naively from the volume sampled itself). A slight bias in the value used in $n_B$ will translate in a bias in the estimation of $\xi$, and thereby, in the estimation of the cross-correlation function. For instance, in a survey with one million galaxies, the variance (7) will be virtually zero, for reasonable binning. However, the clustering within the sample will produce an uncertainty of the order of 1% or more for the estimation of the galaxy density (per luminosity bin). This may be the actual source of the limit on the accuracy by which the bias ratio can be evaluated.

## 5. Discussion

The use of the ratio of (cross-)correlations clearly enables us to obtain the sampling variance for the estimation of bias ratios of distinct populations. One may wonder whether additional information can be obtained from a multi-tracer approach. Having high-quality estimations of the bias is clearly a way to improve the estimations on the average clustering properties of each population and thereby of their combination. However, a simple argument leads to think that not much improvement is to be expected from other statistics: let us assume that we have a single population at hand and split it randomly in two samples *A* and *B*. The relative bias of the two populations can clearly be measured accurately to its value ($=1$), something of no interest in this case, and there is clearly no reason that this will allow us to gain any additional information on other clustering properties. For two distinct populations, with different bias, the knowledge of the bias ratio might help, however, this is likely to be at a limited level.

**Funding:** This research received no external funding.

**Data Availability Statement:** Not applicable.

**Conflicts of Interest:** The author declares no conflict of interest.

## Note

[1] By sampling variance, we refer to the fact that one sample of finite volume is one realization and therefore differs from another realization with the same geometry, tracers, etc., but taken somewhere else. Cosmic variance is achieved when the sample is the largest one possible to build in our observable universe, which is finite. Planck CMB intensity surveys are essentially cosmic variance-limited, at least, for a significant part of the large-scale modes.

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
