# Peer review of "The Role of Cross-Correlations in the Multi-Tracer Area"

_universe, doi:10.3390/universe8090479_

Round 1
Reviewer 1 Report
The author deals with the limitation on cosmological observations coming from the restricted size of the sampled volumes. The author uses a multi-tracer approach to estimate quantities which are insensitive to the noise produced by sampling variance. The method is based on surveys that sample several tracers over overlapping volumes taking advantage of the suppression of the sampling noise in the estimation of the ratio of cross correlations. The paper concludes with a specific example which explains the origin of the suppression of the sampling noise. The content is well organised and the presentation is clear (I think there is only a minor misprint in the discussion section).
The topic is up to date and I suggest publication.
Author Response
Thanks for your comments. The revised version take them into account.
Reviewer 2 Report
The manuscript I reviewed explains an useful technique in an understandable way, but the draft needs to be thoroughly copy-edited as it contains numerous typos and other linguistic issues. I will list the ones I found in the abstract and introduction as examples, but the rest of the draft has similar issues too:
Title: "multi-tracers area" -> "multi-tracer area"
L. 1: "allows to estimate" -> "allows us to estimate"
L. 4: "approach's" -> "approaches"
L. 8: "progresses" -> "progress"
L. 9: "trigerred" -> "triggered"
L. 11: "human made" -> "human-made"
L. 12: " GeV" -> "~GeV" (prevents a line break)
L. 14: "necessary to" -> "necessary for"
L. 15: "within linear regime" -> "within the linear regime"
L. 15: "i.e. their" -> "i.e.\ their" (prevents TeX from inserting sentence-final spacing here)
L. 18: ""low redshift"" -> "``low redshift''" (backticks and apostrophes to be converted by TeX into proper opening and closing quotation marks)
L. 20: "limitations comes" -> either "limitations come" or "limitation comes"
L. 22: "close" -> "closed"
Furthermore:
* The abstract is way too concise, and should be expanded to give a better description of the contents of the manuscript.
* In the second paragraph of Section 3, it would be useful to describe how the FoM is defined, as well as the origin of the acronyms "GCs" and "GCp" (which, by the way, are formatted differently than in Figure 1).
* The proper TeX markup for the angle bracket notation for averages is \left<...\right> (using just <...> results in less-than and greater-than signs, which are too wide, not tall enough, and have too much inside space for this purpose).
Author Response
Thanks for your comments. I have revised the content and tried to respond in this revised version.
Round 2
Reviewer 2 Report
Dear editors,
The revised version of the manuscript looks fine to me and is suitable for publication, except for a few minor language issues:
L. 3 "an information" should be just "information" or "a piece of information" ("information" isn't normally a count noun in English)
L. 5 "ratio of" → "ratio between"
L. 17 "regime i.e." → "regime, i.e."
L. 27 "surveys period" → "survey period" or else "period of ... surveys"
L. 29 "surveys for" → "surveys of"; "CMB which the" → "CMB, whose"
L. 29-31 "the concept of ... projects": this sentence appears to be missing a main verb
L. 34-35 "early identified early"
L. 44-45 "an additional information" → "additional information" or "an additional piece of information" (see above)
L. 48 (in the footnote): "etc. but" → "etc.\ but" (otherwise TeX thinks a sentence ends here and adds too much space); "cosmic variance limited" → "limited by cosmic variance", or at least "cosmic-variance limited"
L. 49 "each tracer have" → "the tracers have" or "each tracer has"
L. 57 "the suing the ration" -- I guess "using the ratios"?
L. 75 "content" → "contain"
L. 77 "non gaussiannity" → "non-Gaussianity" (hyphen, capital G, one n)
Eq. (5) and (6): I suggest writing the index i to the right of the summation symbol rather than underneath it, at least in the numerator (LaTeX should do this automatically, so I suspect the authors used \limits or \displaystyle)
Best regards,
The reviewer
Author Response
Thank you for your careful reading. All the issues you raised have been corrected.